# Recovery from the Neuroparalysis Caused by the *Micrurus nigrocinctus* Venom Is Accelerated by an Agonist of the CXCR4 Receptor

**DOI:** 10.3390/toxins14080531

**Published:** 2022-08-02

**Authors:** Marco Stazi, Federico Fabris, Julián Fernández, Giorgia D’Este, Michela Rigoni, Aram Megighian, José María Gutiérrez, Bruno Lomonte, Cesare Montecucco

**Affiliations:** 1Department of Biomedical Sciences, University of Padova, Via Ugo Bassi 58/B, 35131 Padova, Italy; federico.fabris.5@phd.unipd.it (F.F.); giorgia.deste@studenti.unipd.it (G.D.); rigonimic@gmail.com (M.R.); aram.megighian@gmail.com (A.M.); 2Instituto Clodomiro Picado, Facultad de Microbiología, Universidad de Costa Rica, San José 11501, Costa Rica; julian.fernandezulate@ucr.ac.cr (J.F.); jose.gutierrez@ucr.ac.cr (J.M.G.); bruno.lomonte@ucr.ac.cr (B.L.); 3CIR-Myo, Department of Biomedical Sciences, University of Padova, Via Ugo Bassi 58/B, 35131 Padova, Italy; 4Padova Neuroscience Center, University of Padova, Via Orus2/B, 35129 Padova, Italy; 5CNR Institute of Neuroscience, 35131 Padova, Italy

**Keywords:** Micrurus, snake neurotoxins, presynaptic neurodegeneration, regeneration, CXCR4

## Abstract

Snake envenoming is a major but neglected human disease in tropical and subtropical regions. Among venomous snakes in the Americas, coral snakes of the genus *Micrurus* are particularly dangerous because they cause a peripheral neuroparalysis that can persist for many days or, in severe cases, progress to death. Ventilatory support and the use of snake species-specific antivenoms may prevent death from respiratory paralysis in most cases. However, there is a general consensus that additional and non-expensive treatments that can be delivered even long after the snake bite are needed. Neurotoxic degeneration of peripheral motor neurons activates pro-regenerative intercellular signaling programs, the greatest of which consist of the chemokine CXCL12α, produced by perisynaptic Schwann cells, which act on the CXCR4 receptor expressed on damaged neuronal axons. We recently found that the CXCR4 agonist NUCC-390 promotes axonal growth. Here, we show that the venom of the highly neurotoxic snake *Micrurus nigrocinctus* causes a complete degeneration of motor axon terminals of the soleus muscle, followed by functional regeneration whose time course is greatly accelerated by NUCC-390. These results suggest that NUCC-390 is a potential candidate for treating human patients envenomed by *Micrurus nigrocinctus* as well as other neurotoxic *Micrurus* spp. in order to improve the recovery of normal neuromuscular physiology, thus reducing the mortality and hospital costs of envenoming.

## 1. Introduction

Snakebite envenoming remains a largely neglected tropical disease, despite the fact that it occurs all over the world, especially in tropical and subtropical countries, causing more than 100,000 deaths every year, with an even higher number of victims suffering from physical and psychological sequelae [1,2,3,4,5,6,7,8,9,10]. The venom glands of advanced snakes secrete multicomponent mixtures of proteins, including toxins acting on a range of essential physiological functions. Many Elapidae and some Viperidae snake venoms cause neuromuscular paralysis, which is essential for capturing the prey and may cause potentially fatal respiratory paralysis in humans [11]. The main components of Elapidae snake venoms are (a) phospholipase A_2_ (PLA_2_) enzymes that have evolved to act on the presynaptic motor axon terminals (β-neurotoxins) or the muscle sarcolemma (myotoxins) and (2) neurotoxins of the three-finger toxin family (3FTx) that bind with high affinity the nicotinic acetyl choline receptor (nAChR) located on the post-synaptic muscle membrane (α-neurotoxins) [12]. These neurotoxins cause a blockade in the neuromuscular transmission, which clinically results in flaccid paralysis of various muscles.

The snakes of the genus *Micrurus*, collectively known as coral snakes, belong to the Elapidae family and inhabit the Americas from Southern USA to Argentina, comprising more than 80 species [13]. As in other elapids, the sum of PLA_2_ and 3FTx toxins accounts for 80–100% of the venom proteomes of coral snakes, and on the basis of their proteomic profiles, *Micrurus* venoms have been classified as either PLA_2_-rich or 3FTx-rich [12]. In the case of *Micrurus nigrocinctus,* the medically most important coral snake in Central America which causes potentially fatal envenomings [14,15], their proportion is 48% and 38% of the total venom, respectively [16], thus being a PLA_2_-rich venom. Elapid neurotoxins cause neurological manifestations due to their effects on the nerve terminals of both skeletal and autonomic peripheral neurons, as well as their post-synaptic effect on the AChR. After venom injection, local pain and paresthesia are reported, in addition to other neurotoxic manifestations such as palpebral ptosis, ophthalmoplegia, dysphagia, salivation, and diplopia, followed by a progressive descending flaccid paralysis which, in severe cases, involves respiratory muscles, causing respiratory paralysis that may require mechanically assisted ventilation to prevent death [17,18].

Envenomings by *Micrurus* spp. can be effectively treated by the administration of horse-derived antivenoms [17,18]. However, the difficulties in keeping these snakes alive in captivity and their low venom yield make antivenom production a difficult task, thus generating limited availability in several countries. Moreover, some available antivenoms have limited cross-reactivity. For example, the monospecific *M. nigrocinctus* antivenom manufactured in Costa Rica is generally effective against PLA_2_-rich coral snake venoms but not as effective against 3FTx-rich venoms [13,16]. In addition, there are several species of *Micrurus* for which no effective antivenoms are available. Therefore, aside from widening the scope of neutralization and the production volume of current antivenoms. there is a need to find alternative therapeutic resources that could be used in the treatment of coral snakebite envenomings.

It was recently shown that intoxication of motor axon terminals by different presynaptic neurotoxins induces degeneration of the axon terminals with the formation of an axonal stump expressing the CXCR4 receptor. At the same time, nerve terminal degeneration induces the perisynaptic Schwann cells to express and release the chemokine CXCL12α, which is the specific ligand of the CXCR4 receptor, thus generating an intercellular signaling axis leading to regrowth of the axon and reformation of a functional neuromuscular junction (NMJ) [19]. A CXCR4 chemical agonist, dubbed NUCC-390, was found to act as CXCL12α and to promote axonal growth, effectively shortening the time period necessary to recover from the neuroparalysis induced by α-latrotoxin [20] and by taipan and krait snake venoms, whose major neurotoxic components are potent PLA_2_ presynaptic neurotoxins [21,22]. NUCC-390 is stable and water soluble, and upon i.p. injection, it rapidly redistributes in the body, including the soleus muscle, reaching about 10 nmoles/gram of muscle within 1 hour from injection (unpublished results).

In order to expand these previous observations to coral snake venoms, here, we describe the finding that *M. nigrocinctus* envenoming stimulates the expression of CXCR4 on the motor neuron axon terminal, and this provides the biological basis for the use of NUCC-390, which was found to accelerate 0recovery from the peripheral neuroparalysis induced by this venom. These findings strongly suggest that NUCC-390 might be a novel therapeutic resource to be used in patients bitten by *M. nigrocinctus* and most likely by related *Micrurus* species whose neurotoxic activity is predominantly due to the action of presynaptic PLA_2_ neurotoxins.

## 2. Results

### 2.1. The Degeneration of the Motor Axon Terminals Induced by M. nigrocinctus Venom Is Accompanied by Expression of the CXCR4 Receptor

*M. nigrocinctus* venom is likely to exert its neurotoxic effect by the combined action of pre- and post-synaptically acting PLA_2_ neurotoxins and 3FTxs, respectively [17]. Preliminarily, we tested different doses of *M. nigrocinctus* venom injected in the soleus muscle and analyzed by imaging markers of the presynaptic (VAMP-1) [23] and post-synaptic (α-bungarotoxin (α-BTx)) motor axon (neurofilament (NF)) markers. Figure 1 shows that the injection of 6 µg/kg led to degeneration of the motor axon terminals of the NMJ, which was evident 24 h after venom injection in the soleus muscle and was then followed by a progressive recovery estimated to be about half complete after about 96 h and complete about 168 h. Such a time course of degeneration and regeneration of the presynaptic nerve terminals is similar to those of taipan and krait venoms, as expected because of the similar content of their presynaptic PLA_2_ neurotoxins. In fact, the neurodegeneration associated with the neuroparalysis of the NMJ resulted from the action of the PLA_2_ β-neurotoxins acting presynaptically and leading to loss of the end plate while leaving intact the post synapse. Previous observations indicated that 3FTx neurotoxins were also present in these venoms and bound with high affinity to AChR, and the α-neurotoxin-AChR complex was removed from the membrane within a few days in the denervated rodent neuromuscular synapse, while the unbound clustered AChR was retained longer on the folded muscular membrane [24].

Figure 2 shows that at 24 h after venom injection, CXCR4 begins to be detectable, and its expression becomes evident at 48 h, remaining elevated until NMJ regeneration is accomplished. Corresponding quantitation and statistical treatment of the imaging data are given in the panels on the right-hand side of the figure. This finding is important per se and because it provides the basis for testing the effect of the CXCR4 agonist NUCC-390 with respect to a possible acceleration of the recovery of NMJ function, thus decreasing the duration of the neuroparalysis caused by this venom, as was previously found for taipan and krait venoms [21,22].

### 2.2. NUCC-390 Stimulates the Recovery of the NMJ Morphology and Function after Degeneration of the Motor Axon Terminals Induced by M. nigrocinctus Venom

Electrophysiological and imaging techniques were used here to estimate the effect of NUCC-390 in promoting the functional recovery of NMJ function after venom-induced neuroparalysis. Experiments were performed on the soleus muscle, which was used as a test muscle among the hind limb muscles of mice. A scheme of the technique employed and the workflow of the experiment are depicted in Figure 3 in panels A and B, respectively. The evoked junctional end plate potential (EJP) at zero time and after envenoming of the mice treated or not treated with daily injections of NUCC-390 was measured. We found that this CXCR4 agonist induced a more extensive recovery of function at 96 h, a time when regeneration was still under way, thus representing an appropriate time point to detect the effect of NUCC-390 on the recovery of functionality of the NMJ (Figure 3C). This electrophysiological measurement was paralleled by morphological analysis of the expression of the presynaptic membrane markers VAMP1 and neurofilaments performed on the same muscle analyzed by EJP recordings (Figure 3D). Indeed, this figure shows a higher number of re-innervated NMJs in NUCC-390-treated muscles compared with the untreated samples, as documented by the higher VAMP-1 signal (*green*) that better overlapped with α-BTx staining of the post-synaptic muscle membrane (*red*). In addition, the “bretzel” shape of the mouse NMJ was more accomplished and evident. Quantification of the number of normal appearing synapses with or without NUCC-390 treatment is reported in panel E of the same figure.

### 2.3. NUCC-390 Stimulates Rehabilitation of the Neuromuscular Junction after Degeneration of the Axon Terminals Induced by M. nigrocinctus Venom

Measurement of the EJP provides an estimation of the functional state of single fibers, although the data of Figure 3 report the outcome of a high fiber number. To strengthen our data, we also analyzed the compound muscular action potential (CMAP), which allowed an evaluation of the recovery of function of the entire muscle, thus providing a more representative scenario of clinically relevant observations (a scheme of the workflow is depicted in Figure 4A). Panels B and C of Figure 4 show that the recovery of CMAP was increased by NUCC-390. Indeed, after 4 days, the value of recovered CMAP following paralysis was much higher in the mice treated with NUCC-390, indicating that stimulation of the neuronal receptor CXCR4 was effective in expediting recovery from paralysis. Moreover, representative CMAP traces (panel C) from the NUCC-390-treated animals display the physiological control shape (two peaks because they were extracellularly recorded) at variance from the vehicle traces that were polyphasic, indicative of an ongoing denervation-reinnervation process. In fact, this process altered the physiological activation of the motor unit at the basis of the biphasic nature of CMAPs. This led to an asynchronous activation of the muscle motor units and therefore the observed “vehicle” traces. This conclusion was further supported by imaging analysis of the expression of the presynaptic markers VAMP-1 and neurofilaments performed on similarly envenomed mice. Accordingly, Figure 4D shows a higher number of re-innervated NMJs in NUCC-390-treated muscles compared with the vehicle-treated samples. These data reinforce the conclusion that NUCC-390 is to be considered a novel therapeutic candidate to be tested for its property of accelerating nerve structural and functional recovery from the peripheral neurodegeneration and paralysis caused by *M. nigrocinctus* envenoming.

### 2.4. NUCC-390 Stimulates Restoration of Respiratory Function Impaired by M. nigrocinctus Venom

A major clinical manifestation of envenoming by *Micrurus* snakes is the respiratory deficit that may be so severe as to cause death. Therefore, we attempted to test the effect of NUCC-390 on the recovery of respiratory function in envenomed mice. This physiological parameter was assessed using an indirect but very sensitive and minimally invasive test. A probe connected to a pressure sensor was placed inside the esophagus at the mediastinum level in anesthetized mice. The recorded signal was characterized by asymmetric peaks with a frequency corresponding to the events of lung ventilation. The peak area may be taken as an estimate of the volume of air inspired, which is directly related to pressure variations resulting from the activity of muscles involved in breathing. The i.p. injection of *M. nigrocinctus* venom caused a decrease in both the frequency and extent of ventilation, with changes in the shape and dimension of each peak (Figure 5A). The NUCC-390-treated animals already showed an increase in peak area compared with the vehicle-treated mice after 24 h, and this difference increased 4 days after envenoming (panels A and B of Figure 5). Indeed, at 96 h, the NUCC-390-treated mice showed a complete recovery in lung ventilation, a result that was reached by the control animals only after 1 week.

Quantitative analysis performed on a group of mice showed that after an initial similar decline due to envenoming (24-h time point in the graph in Figure 5B), NUCC-390 accelerated the recovery of ventilation. This functional recovery was accompanied by restoration of the structure of the diaphragm NMJs, as shown in panels C and D of Figure 5.

## 3. Discussion

We report here experimental evidence that the peripheral neuroparalysis caused by the neurotoxic venom of a coral snake species, *M. nigrocinctus*, is associated with degeneration of the motor neuron end plate with prolonged expression of the CXCR4 receptor on the axon terminal. In addition, we found and documented with different experimental techniques that NUCC-390, a low molecular mass CXCR4 agonist, accelerated the recovery of the structure and function of the mouse NMJs with respect to the untreated controls. An additional major result is that NUCC-390 induces a faster recovery of the respiratory function, a severe consequence of envenoming by neurotoxic snakes. This is very relevant for envenomated patients that frequently require mechanically supported lung ventilation, which is often not available in many areas of low-income countries where snakebite envenoming is more frequent. Owing to the existing difficulties in the availability of *Micrurus* antivenoms in the Americas, this alternative therapeutic approach might be helpful when patients require mechanical ventilation, since it speeds up the recovery process.

*Micrurus* venoms act both at the pre- and post-synaptic levels because they contain as major components presynaptically acting PLA_2_s and post-synaptically acting 3FTxs [12,13,16,25,26,27,28]. The post-synaptic blockade of the AChR ion channel and, consequently, of muscle fiber contraction does not cause degeneration of the nerve terminal or the muscle fiber, and the 3FTx-AChR is removed from the muscle membrane within a few days in the denervated NMJ [24]. On the contrary, the action of the presynaptic PLA_2_ snake neurotoxins causes membrane damage followed by massive entry of Ca^2+^ into the terminal, with the consequent inhibition of mitochondrial function and ATP generation together with Ca^2+^-induced hydrolytic degradation of proteins, lipids, and nucleic acids [29,30,31,32]. The spider α-latrotoxin causes nerve terminal degeneration similar to that induced by some neurotoxic snake venoms. The primary cause of this degeneration is the overload of nerve terminals, with Ca^2+^ flowing along the concentration gradient from the outside to the inside of the cell. Such degeneration is an acute phenomenon that takes place within a few hours [33]. We previously showed that regeneration follows α-latrotoxin-induced degeneration. A central event of regeneration is the expression of CXCR4 on the axon terminal. The natural ligand of CXCR4 is the chemokine CXCL12α produced by activated perisynaptic Schwann cells, which stimulates axonal growth [19]. A chemical CXCR4 agonist has been discovered and dubbed NUCC-390 [34]. This agonist is not toxic and stimulates the recovery of function similar to CXCL12α [20]. How exactly NUCC-390 acts on CXCR4 and which intracellular signaling pathway is activated in motor neurons remain to be investigated in detail.

NUCC-390 is the first drug found to be capable of speeding up the recovery of function of the neuromuscular apparatus after experimental neuroparalytic snake envenoming. This is particularly important in humans, as recovery of the physiological function of the neuromuscular apparatus after envenoming with a neurotoxic snake venom acting predominantly on the presynapse requires more than a month, as judged by measurement of the CMAP [35]. In the case of *Micrurus* spp. venoms, it is likely that this would apply to all venoms containing presynaptically acting PLA_2_s, as shown here for *M. nigrocinctus*. NUCC-390 is expected to decrease the recovery time after peripheral neuroparalytic envenoming in humans as well, and this would result in shortening hospital stays with a parallel decrease in healthcare costs and a less drastic impact on the life of the affected person. We have designed a simple and effective synthesis of NUCC-390, and its production should not be expensive. The drug has not been used in clinical trials yet, but the present findings indicate that all the preliminary conditions to set up a preclinical assessment of the efficacy of NUCC-390 with other neurotoxic snake venoms should be pursued, ideally followed by clinical trials in neurotoxic snakebite envenomings.

## 4. Materials and Methods

### 4.1. Antibodies, Reagents, and Toxins

The following primary antibodies were employed: anti-VAMP1 (1:200, generated as described in [23]), anti-CXCR4 (Abcam, cat. Ab 124824, 1:400, Cambridge, UK), and anti-Neurofilament (Abcam, cat. Ab 4680, 1:800). The α-bungarotoxin (α-BTx) (cat. B35451, 1:200) and Alexa-conjugated secondary antibodies (1:200) were from Life Technologies. Unless otherwise stated, all other reagents were from Sigma. NUCC-390 was synthesized as previously described in [20].

The *M. nigrocinctus* venom was a pool of more than 50 specimens collected in the central and Pacific regions of Costa Rica and kept at the Serpentarium of Instituto Clodomiro Picado (University of Costa Rica). Once obtained, the venom was freeze-dried and stored at −20 °C.

### 4.2. Evoked Junctional Potentials (EJPs)

Mice were intramuscularly injected in the hind limb with 6 μg/kg *M. nigrocinctus* venom (in 15 μL physiological solution with 0.2% gelatin) followed by daily intramuscular administration of NUCC-390 (3.2 mg/kg in 20 μL physiological solution with 0.2% gelatin) or a vehicle only. Ninety-six hours after venom injection, the mice were killed, and their soleus muscles were quickly dissected, processed, and analyzed as previously described in [21].

### 4.3. Compound Muscle Action Potential (CMAP)

The mice were injected intramuscularly in the hind limb with 6 μg/kg *M. nigrocinctus* venom (diluted in 15 μL physiological solution containing 0.2% gelatin), followed by daily intraperitoneal (i.p.) administration of NUCC-390 (3.2 mg/kg in 20 μL physiological solution with 0.2% gelatin) or a vehicle only. Then, 96 h later, CMAP analysis was performed as previously described in [21].

### 4.4. Ventilation (IVI) Recordings

Following general anesthesia, the animals were left in their cages to relax for 10 min. For each mouse, recordings were performed before (t0) and 24, 96, or 168 h after intoxication with 6 µg/kg *M. nigrocinctus* venom (diluted in 40 μL physiological solution plus 0.2% gelatin). The mice were injected intraperitoneally with 3.2 mg/kg NUCC-390 (diluted in 30 μL physiological solution plus 0.2% gelatin) or a vehicle only daily. A 20 ga × 38-mm plastic feeding tube (Instech Laboratories, Inc., Plymouth Meeting, PA 19462, USA) connected to a pressure sensor (Honeywell, 142PC01D) was carefully introduced into the oral cavities of the mice and placed in the lower third of the esophagus at the level of the mediastinum. The animals were laid on their left sides on a pre-warmed surface to record their mediastinic pressure variations, which were used to infer animal ventilation. Traces were recorded, amplified, and digitized with WinEDR V3.4.6 software (Strathclyde University, Glasgow, Scotland). Stored data were analyzed using Clampfit software (Axon, New York, NY, USA). For each animal, 120 epochs were recorded, and 20 epochs were analyzed at each time point. The IVI was then calculated as the product of the mean area of the peaks (VV × ms) with the number of peaks within 20 sec and represented in the graphs as the percentage of recovery of the animals compared with their own initial records (t0) [22]. By measuring the esophageal pressure variations, which reflected the intrapleural pressure variations, we indirectly estimated the inspired volume of air.

### 4.5. Immunohistochemistry

The anesthetized mice which were locally injected with 6 µg/kg of *M. nigrocinctus* venom were injected within the soleus muscle. After being sacrificed, the muscles were dissected at different time points and stained as previously described in [21].

### 4.6. Statistical Analysis

The experiments were conducted blindly, while the sample sizes were based on papers previously published by our group. Here, *n* = 4 mice/group were used for EJP and CMAP analysis. Data were displayed as histograms and expressed as means ± SEM. Statistical analyses were performed with GraphPad Prism software, using an unpaired Student’s *t*-test or one-way analysis of variance (ANOVA) with Tukey’s post-test when more than two experimental conditions were compared to each other. Differences among the groups were defined as significant when *p* < 0.05 (*), *p* < 0.01 (**), *p* < 0.001 (***), or *p* < 0.0001 (****).

### 4.7. Ethical Statement

General anesthesia and analgesia were applied in all in vivo experiments, where paralysis was confined to one hind limb only without affecting food or water intake. Regarding the immunofluorescence experiments of the CXCR4 receptor, Dr. W.B. Macklin (Aurora, Colorado) kindly provided C57BL/6 plp-GFP mice. CD1 mice (25–30 g) were used for EJP, CMAP, and imaging analysis. All in vivo experiments were performed following national laws and policies (D.L. n. 26, 14 March 2014) and the guidelines of the European Community Council Directive (2010/63/EU) and approved by the ethical committee and the animal welfare coordinator of the OPBA of the University of Padova. All procedures are specified in the projects approved by the Italian Health Ministery with authorization n° 359/2015 (approved 11 May 2015).

## Figures and Tables

**Figure 1 toxins-14-00531-f001:**
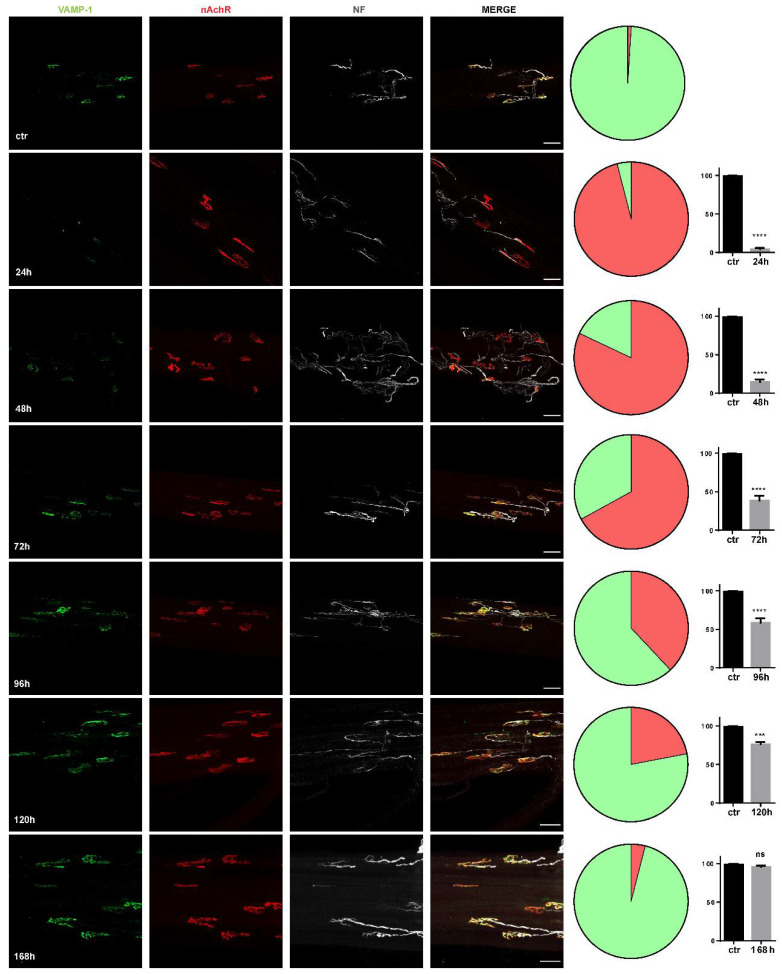
*M. nigrocinctus* venom injection in the soleus muscle induced an acute and reversible degeneration of MAT at the NMJ. The degeneration induced by the injection of *M. nigrocintus* total venom (6 µg/kg) was monitored by immunostaining. The motor axon terminal (MAT) is identified by VAMP-1 immunostaining (*green*), post-synaptic nAChRs is identified by fluorescent α-BTx (*red*), and the axon terminal is identified by neurofilament (NF) staining (*white*). Scale bars: 50 µm. The pie chart shows the extent of neurodegeneration (*red* part) compared with intact NMJ (*green*), followed by their respective quantification and statistics (paired *t* test, *n*= 4). *** *p* < 0.001. **** *p* < 0.0001, ns (not significant).

**Figure 2 toxins-14-00531-f002:**
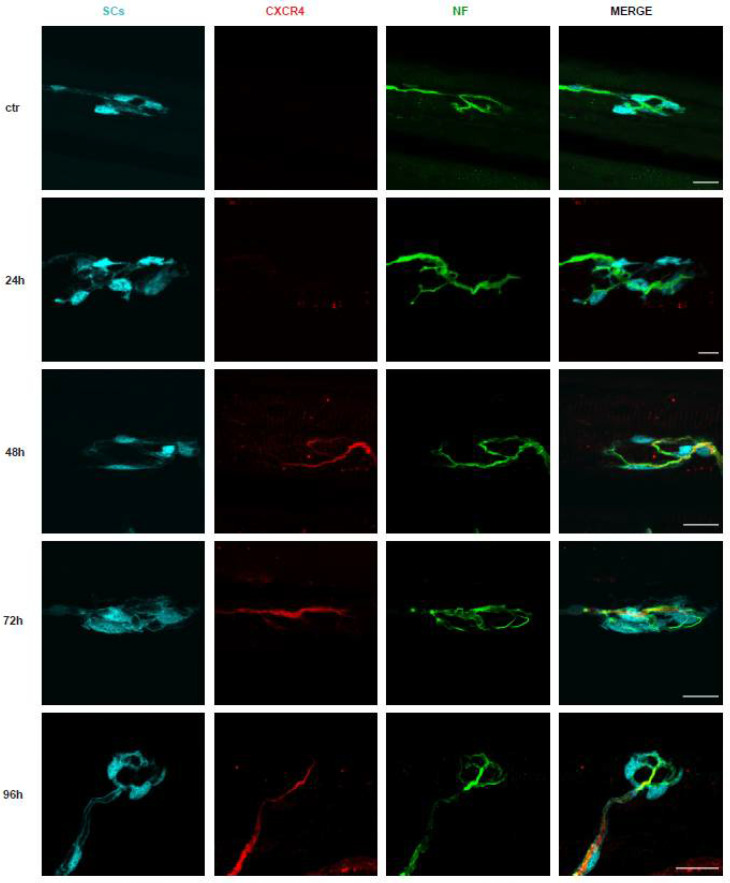
CXCR4 receptor is expressed in neuronal axons after *M. nigrocinctus* venom injection. CXCR4 staining (*red*) at soleus NMJs in controls (upper panels) and after *M. nigrocintus* venom injection (lower panels). Schwann cells (SCs) were GFP-positive (*cyan*), and the axon terminal was identified by NF staining (*green*). Scale bars: 10 µm.

**Figure 3 toxins-14-00531-f003:**
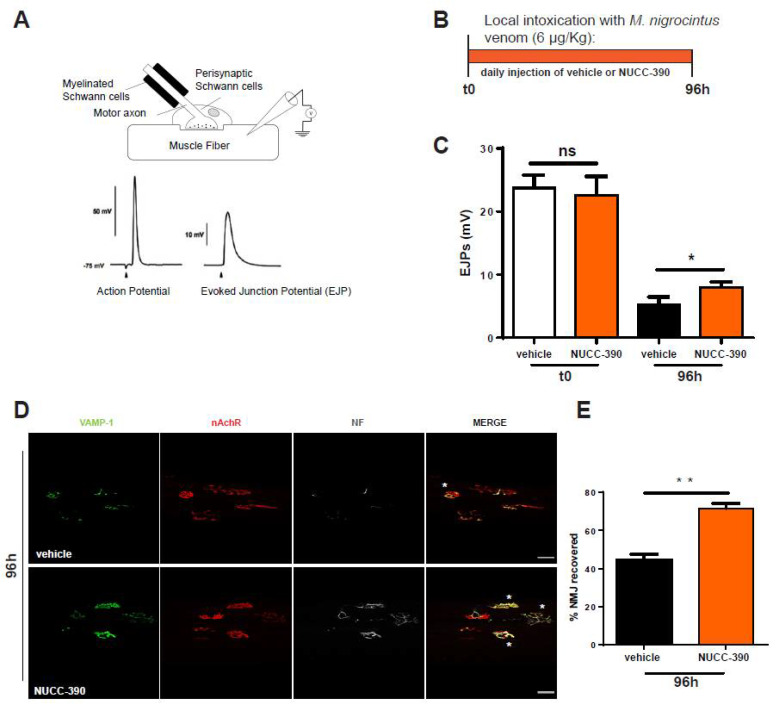
NUCC-390 promoted NMJ recovery of function after *M. nigrocinctus* venom intoxication. (**A**) Schematic representation of the technique of measurement of the evoked junctional potentials (EJPs). (**B**) Temporal scheme of the administration of *M. nigrocinctus* venom and NUCC-390 administration. (**C**) EJPs of soleus muscles 96 h post injection of *M. nigrocinctus* venom in the hind limb with or without daily NUCC-390 administrations. Each bar represents the mean of the EJP amplitude ± SEM from *n* = 4 (number of analyzed fibers = 12, one-way ANOVA with Tukey’s multiple comparison test; ns = not significant). * *p* < 0.05. (**D**) Representative immunostaining of NMJs performed on the same muscles used for EJP measurements. (**E**) Quantification of NMJs from (**D**) (paired *t* test, *n* = 4). ** *p* < 0.01. MAT is identified by VAMP-1 immunostaining (*green*), post-synaptic nAChRs are identified by fluorescent α-BTx (*red*), and the axon terminal is identified by NF staining (*white*). White asterisks indicate still-degenerated NMJs. Scale bars: 50 µm.

**Figure 4 toxins-14-00531-f004:**
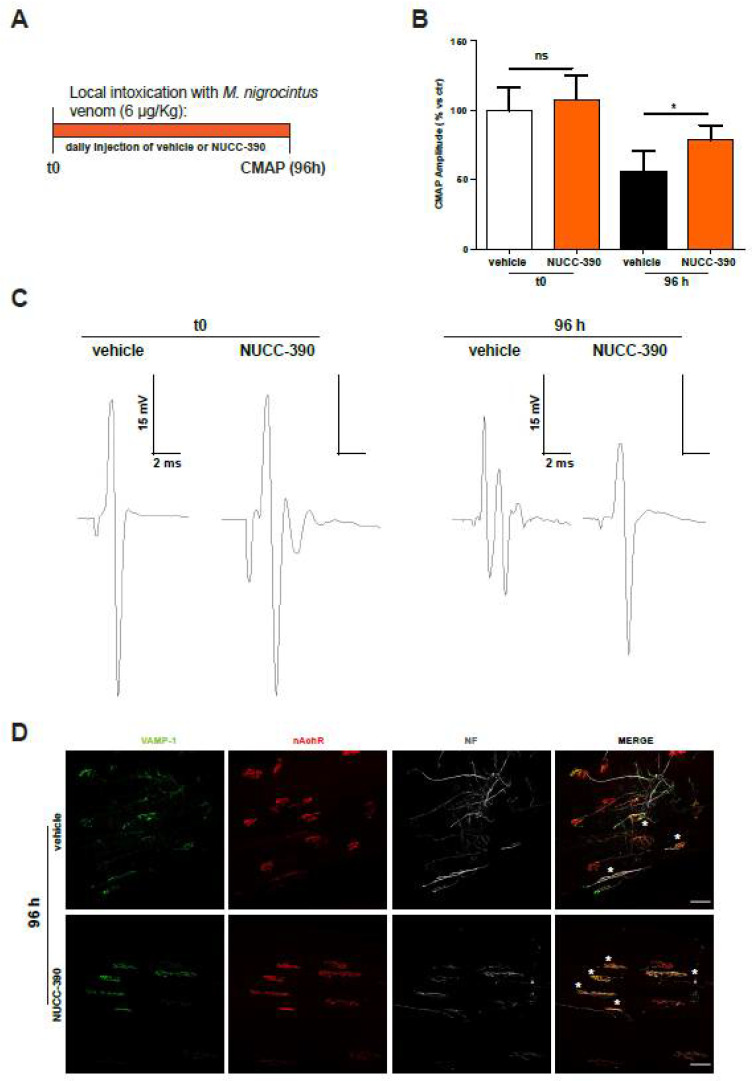
NUCC-390 increased the recovery of compound muscle action potential from paralysis induced by *M. nigrocinctus* venom. (**A**) Temporal scheme of the intoxication of the soleus muscle induced by local injection of *M. nigrocinctus* venom with or without NUCC-390’s daily administration. (**B**) CMAP values recorded for gastrocnemius muscles 96 h after envenomation with *M. nigrocinctus* venom with or without NUCC-390’s daily local administration. Data are expressed as a ratio between envenomed and control animals CMAP amplitude ± SEM (*n* = 5, one-way ANOVA with Tukey’s multiple comparison test; ns = not significant) * *p* < 0.05. (**C**) Representative traces of CMAP analysis of control animals (**left**) and 96 h after envenomation (**right**). (**D**) Representative immunostaining of intoxicated NMJs performed on the same muscles used for CMAP analysis. Motor neuron axon terminals are identified by VAMP-1 immunostaining in *green*, post-synaptic nAChRs are identified by fluorescent α-BTx (*red*), and the axon is identified by NF staining in *white*. Regenerated NMJs are indicated by white asterisks. Scale bars: 50 μm.

**Figure 5 toxins-14-00531-f005:**
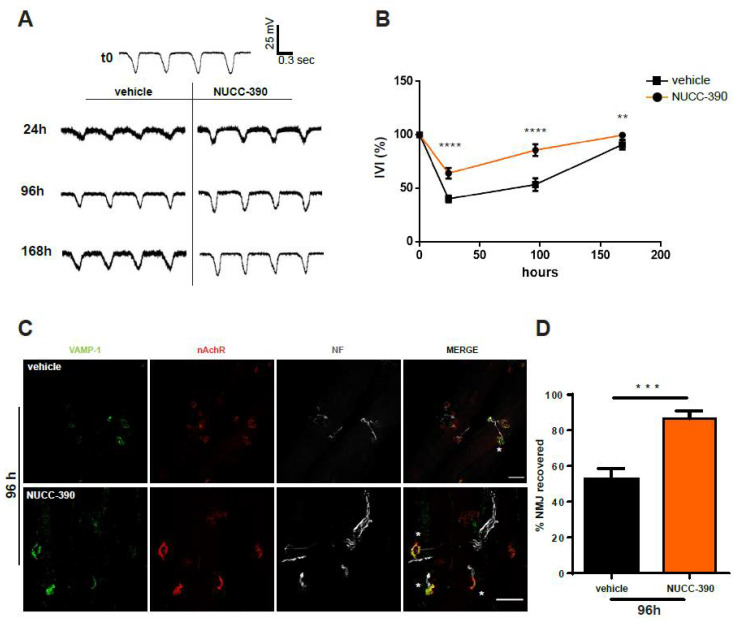
NUCC-390 increased lung ventilation recovery in mice intoxicated with *M. nigrocinctus* venom. (**A**) Representative traces of mediastinum pressure variations in untreated mice (t0) or 24 h, 96 h, or 168 h after *M. nigrocinctus* administration (6 μg/kg) with NUCC-390 or vehicle i.p. daily injection (3.2 mg/kg in 40 μL physiological solution containing 0.2% gelatin). (**B**) The inferred ventilation index (IVI) was estimated as measurement of the peak area of 20 consecutive events of the traces (±SEM) obtained from a group of mice envenomed with *M. nigrocinctus* venom with (orange trace) or without (black trace) NUCC-390 daily administration (one-way ANOVA with Tukey’s multiple comparison test, *n* = 4). ** *p* < 0.01. **** *p* < 0.0001. (**C**) Representative immunostaining of envenomed NMJ 96 h after envenoming at the level of the diaphragm muscle daily injected i.p. with either vehicle (upper panels) or NUCC-390 (lower panels). Motor axon terminals are identified by VAMP-1 immunostaining (*green*), post-synaptic muscle membrane is identified by nAChR staining with fluorescent α-BTx (*red*), and axon is identified by NF staining (*white*). Scale bars: 50 μm. (**D**) Quantification of recovered NMJ stained in panel C (paired *t* test, *n* = 4). *** *p* < 0.001.

## Data Availability

The data presented in this study are available in this article.

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
