# Peer review of "Recovery from the Neuroparalysis Caused by the *Micrurus nigrocinctus* Venom Is Accelerated by an Agonist of the CXCR4 Receptor"

_toxins, 2022, doi:10.3390/toxins14080531_

Round 1

Reviewer 1 Report

1.      The manuscript does a good job of introducing the topic and the significance of the research.

2.      The paper is concise, singularly focused and the experimentation and subsequent results supports the authors conclusions.

3.      A significant amount of in-situ fluorescence experiments is used to support the loss/gain of synaptic development. However, there is no quantitation or tables to support the apparent observations. Can you put together a summary/table that adds significance to these experiments?

4.      A table showing the numerical magnitude and statistical treatments need to be inserted for data that is shown in Figure 3C, 3E, 4B, 5B and 5D.

5.      Can you provide a better explanation to the reader reading the polyphasic and non-traditional profile of the action potentials that are presented in Figure 4C? This is significant data and will not be easily understood by the general population of this manuscript.

6.      If the drug (NUCC-390) is delivered i.p., how do you know that the drug is reaching the target? Do you have any mass spectrometry studies to support the ADME of this drug? Since this drug has been shown previously to increase the rate of regeneration, a figure must be added to show aspects of NUCC-390 bioavailability.

7.      Line 68-78: This paper is premised on the data findings from a similar group pf authors; do you have any other sources that support this NUCC-390 augmentation of regeneration?

8.      Line 279: Is the freeze-dried venom actually stored at 230C; why is the venom not stored in freezer?

9.      Lastly, please provide more detail in the Discussion of the proposed mechanism of action/pathway that NUCC-390 induces to promotes the regeneration of axon terminals.

Reviewer 2 Report

In the manuscript “The recovery from the neuroparalysis caused by the Micrurus nigrocinctus venom is accelerated by an agonist of the CXCR4 3 receptor”, the authors investigated the role of CXCR4 agonist NUCC-390 in Micrurus nigrocinctus venom-induced presynaptic neurodegeneration. NUCC-390 has been studied for motor axon regeneration and Taipan snake envenomation. Here the authors further study the role of NUCC-390 in another snake species, the coral snake. Although I found this article interesting, I have some commons, as listed below: 

Major comments:

All the data showed that NUCC-390 is able to restore the function of EJPs, CAMP, and IVI (Venom-treated groups with/without NUCC-390), the author did not show the data between no venom vs venom treatment except 4B. I think it is important to know the degree of degeneration after venom envenoming.

Fig 1 shows that VAMP-1 and nAchR were induced by snake venom. And Figure 3 shows that NUCC-390 also induces both protein expression. So what do these proteins function here?

Minor comments:

In Fig 2 legend, it is confusing about the statement “Schwann cells (SCs) are GFP-positive (cyan)”. Are they green or cyan?

What is MAT in Fig 1 legend?

Clear labeling should be included in the figures. Figure 3-5, the authors should indicate that they are all under venom envenoming. Fig 4, should control be 0 h, vs 96 h? And is it statistic significant in the vehicle group between control and 96 h?

Reviewer 3 Report

The manuscript “The recovery from the neuroparalysis caused by the Micrurus nigrocinctus venom is accelerated by an agonist of the CXCR4 receptor” examines the use of NUCC-390, a CXCR4 agonist, as a potential small molecule candidate to treat human patients envenomed by M. nigrocinctus. This work is of importance considering antivenoms for coral snakes are difficult to manufacture and for some species antivenom is not currently available. The results presented here demonstrate NUCC-390 improves the recovery of normal neuromuscular physiology after M. nigrocinctus envenoming. Experimental conclusions are well supported with multiple complementary approaches to demonstrate the recovery of neuromuscular junctions after paralysis induced by M. nigrocinctus venom.

Major comments:

Is NUCC-390 an available drug in low- and middle-income countries? Is it expensive? Is this an approved drug? Has it been used in humans? Please address this in the discussion.

Minor comments:

Figure 1: Abbreviation MAT is not defined – same in other figure legends.

Figure 4: Legend mentions that regenerated NMJ are circled in white, but it looks like they have asterisks.

Figure 5: Asterisks are present in this figure as well, but not explained in the legend.

Round 2

Reviewer 1 Report

Thank you for all your responses. These are good studies and I look forward to the work that is coming from your collaborations.

Author Response

we warmly thank the Reviewer for the appreciation of our work

Reviewer 2 Report

All concerns were addressed.

Author Response

we warmly thank the Reviewer for the positive answer